# Deep and continuous sedation until death in the French overseas departments

Sophie Pennec[1,2*], Hélèna Briand[3], Vincent Guion[4], Adrien Evin[5,6]

1 Institut National d'Études Démographiques (INED), Aubervilliers, France, 2 School of Demography, Australian National University (ANU), Acton, Australian Capital Territory, Australia, 3 Nantes Université, Nantes, France, 4 Cabinet médical, Saint-Chély-d'Apcher, France, 5 Nantes Université, Tours Université, Centre Hospitalier Universitaire de Nantes, Centre Hospitalier Universitaire de Tours, Institut National de la Santé et de la Recherche Médicale (INSERM), Methods in patients-centered outcomes and Health Research (SPHERE), Nantes, France, 6 Nantes Université, Centre Hospitalier Universitaire de Nantes, Service Interdisciplinaire Douleur, Soins Palliatifs et de Support, Médecine intégrative, Nantes, France

* pennec@ined.fr

## Abstract

### Background

Continuous deep sedation until death (CDSUD) has been a legally recognised end-of-life right in France since 2016. Data on implementation remain limited, particularly in the French overseas departments, which share the same legal framework as mainland France but present a more rapidly ageing population. This study describes the characteristics of patients who received CDSUD, the profiles of the physicians involved, the modes of implementation, and the emotional impact of CDSUD on physicians.

### Method

We conducted a retrospective, questionnaire-based survey among physicians who certified deaths in four French overseas departments. The study population comprised a random sample of adult deaths between March 2020 and February 2021. CDSUD cases were identified through a multidisciplinary review process.

### Findings

Of 1,082 informed non-sudden deaths, CDSUD was implemented in 128 cases (11.8%), mainly in hospitals, among younger patients without cognitive impairment. midazolam or benzodiazepine were used in 78.8% of cases, often combined with morphine. A colleague was consulted in 80% of cases, though formal collegiality was not systematic. Physicians mainly aimed to relieve unbearable suffering. They acknowledged possible hastening of death, although this was rarely their intention. The emotional impact on physicians was similar, whether or not CDSUD was performed.

**Data availability statement:** Data cannot be shared freely because the department of residence/death can be identified from our dataset and that is considered identifying information by the French liberty act (CNIL). Data can be accessed by researchers affiliated to a research/educational institution, upon submission of a research project through the data progedo plateform (https://data.progedo.fr/). The European General Data Protection Regulation (GDPR) applies.

**Funding:** SP received grants to support the research by the Caisse nationale de solidarité pour l'autonomie (CNSA), as part of the call for projects launched by IReSP (project IReSP-17-Hand8-16) ; by the Fondation de France as part of the call for project "Soigner, soulager, accompagner (2017)". The Institut national d'études démographiques (INED) was involved in the data collection through its Survey department. https://iresp.net/, https://www.fondationdefrance.org/fr/, https://www.ined.fr No funding bodies was involved in the contents of this paper.

**Competing interests:** The authors have declared that no competing interests exist.

## Interpretation

This first population-based study of CDSUD in the French overseas departments highlights the need to improve access to this practice, particularly for older adults and those receiving end-of-life care at home. Further prospective and qualitative research is needed.

## Introduction

### Continuous deep sedation until death: a specific sedation practice

Sedation practices are a component of palliative care, particularly for the management of refractory pain in patients with limited life expectancy [1]. Patients' right to access continuous deep sedation until death (CDSUD) was enshrined in French law in 2016 [2]. France remains the only country with explicit legal provisions for CDSUD [3].

In France, CDSUD can be implemented at the patient's request in two circumstances 1) "if a patient presents refractory symptoms and a serious and incurable condition that is life-threatening in the short term"; 2) "if a patient with a serious and incurable condition decides to discontinue treatment, and this decision has life-threatening consequences in the short term and is likely to result in unbearable suffering". For patients who cannot express their wishes, CDSUD is implemented if the physician discontinues life-sustaining treatment to avoid unreasonable intrusive intervention, unless the patient has objected to this in his/her advance directive" (A document written by an individual expressing their wishes to pursue, limit, discontinue or refuse certain treatments or medical procedures in anticipation of the day when they will no longer be able to make these decisions for themselves) [4].

The French National Health Authority published guidelines in 2018 to assist decision makers with the legal and therapeutic procedures involved [4]. The implementation of CDSUD must be decided during a "collegial procedure" defined as a dialogue between all professionals involved in the patient's care and an outside physician outside called on as a consultant to assess globally and medical the situation the patient's conditions, and record the decisions taken in the patient's medical records. Intravenous midazolam is the recommended first-line medication, titrated to achieve a sedation depth of –4 to –5 on the Richmond Agitation-Sedation Scale or –5 on the Rudkin scale. Sedation must be prescribed in association with analgesia, comfort care, and the withdrawal of life-sustaining treatments, including artificial hydration and nutrition. These national guidelines are in accordance with those proposed by the European Association for Palliative Care (EAPC) revised in 2024 [1].

### Few studies worldwide on the reality of CDSUD practices

Published international data on CDSUD remain limited [5,6]. Studies on decision-making processes and therapeutic practices in France are based on very small patient samples (22 [7], 51 [8]). A French study conducted in 2025 suggests considerable variability in the prevalence of sedation, as well as in its implementation [9]. Available data suggest that the decision-making process is not systematically

recorded in medical records (60% to 83% of cases) and that midazolam is as the most commonly used sedative drug [7,8,10,11]. Data on patients, indications, and prescribing physicians remain scarce. No data was found to describe how sedation impacts physicians in terms of their emotions and their perceptions of their patient's end of life [12].

### An end-of-life study in overseas France to examine this practice

Available data on French practices focus on the situation in mainland France before 2016, and the situation in overseas France remains underreported [13]. Studies have been carried out in various European countries, as well as in mainland France, but before the law on CDSUD came into force [13,14]. The French overseas departments operate under the same legal framework as mainland France and exhibit a similar distribution of causes of death. Over recent years, cancer has emerged as the leading cause of mortality, although cardiovascular disease persists at a high level—indeed higher than in mainland France [15].Their populations are ageing more rapidly and the share of home deaths is higher than in mainland France, making it possible to envisage how practices are liable to change over the near future in mainland France [16,17].

This study describes the characteristics of patients who received CDSUD in overseas France from March 2020 to February 2021, the characteristics of the physicians involved and the modes of CDSUD implementation. We also examine physicians' perceptions of end of life and their emotional response to sedation.

## Materials and methods

### Study design

This ancillary study on the end of life in overseas France (FDVDOM) is based on a survey of physicians conducted between September 2020 and July 2022. It concerns decedents for whom the respondents signed a medical certificate of death between March 2020 and February 2021. The entire research protocol is available on open access [18].

### Ethics statement

The protocol collected testimonies from physicians on the end of life of patients who died during the study period.

The Comité d'Expertise pour les Recherches, les Études et les Évaluations dans le domaine de la Santé [National Expert Committee for Research, Studies and Evaluations in the Field of Health], approved the methodology used, waived the need for consent and waived the requirement of medical confidentiality for research purposes (approval in March 2018).

Data were processed in compliance with the European General Data Protection Regulation (GDPR) as well as national legislation (Data Protection Act). Authorization request No. 918091 was submitted to the Commission Nationale Informatique et Libertés (CNIL), which issued a favourable decision (DR-2018–102).

This study adheres to the ethical principles of the Declaration of Helsinki, particularly in terms of scientific rigour and respect for human dignity. The data used pertain to deceased individuals and were processed in accordance with applicable regulations and the recommendations of the ethics committee.

The dataset is anonymous. The questions on the profiles of physicians and deceased persons included in the questionnaire were designed to ensure that the physician-deceased person pairs could not be identified via the categories used (e.g.: broad categories for age, doctor's specialty).

### Study population

All deaths of persons aged 18 years and over occurring between March 2020 and February 2021 in the French overseas departments (Réunion, Guadeloupe, Martinique, and French Guiana) were included.

Deaths for which the death certificate was signed by a forensic pathologist, or an on-call medical team were excluded. If a physician had certified multiple deaths, they were asked to complete a maximum of four questionnaires per survey wave (except for department heads and coordinating physicians in nursing homes, who could be contacted to forward the questionnaire to their colleagues).

## Data collection

Medical death certificates were obtained from the local health agency (ARS) in La Reunion and from the Centre for the epidemiology of medical causes of death (CépiDc –INSERM) for the other departments. These certificates were used to obtain the contact details of the certifying physicians and to collect basic information on the deceased persons, such as date of birth, date of death, sex, and place of death. No medical information, such as cause of death, was collected. The French Institute for Demographic Studies (INED) mailed paper questionnaires to the physicians who had signed these medical death certificates [18].

Physicians were asked to return the completed questionnaire by post. Each questionnaire was placed in a blank envelope, which in turn was placed in an envelope with an identifier. The trusted third party who received the envelopes opened them and placed the unidentifiable blank envelopes in a box so that they could be mixed to ensure anonymity. The identifier on the envelope ensured that physicians who had replied were not contacted again by INED. Questionnaire responses were then entered into a software program by this trusted third party under strict conditions of anonymity.

Questionnaires were sent in three successive waves to limit recall bias: from March to June 2020, from July to October 2020, and from November 2020 to February 2021. Due to a technical issue, the third wave had to be cancelled in French Guiana. Up to three reminders were sent by post or email and one phone call was made if necessary to maximize the response rate.

## Questionnaire and variables

The questionnaire, accompanied by an explanatory booklet presenting the study, consisted of 41 single or multiple-choice questions, divided into six sections: physician's characteristics, patient's characteristics, end-of-life care and treatments, end-of-life medical decisions, reported final medical act, and persons caring for the patient (S1 and S2 Files). The questionnaire design was based on that of previous studies using the same methodology. It was adapted following a field observatory mission and validated by a multidisciplinary scientific committee (sociologist, member of the ethics committee of the national board of physicians, cause-of-death statisticians, policy maker from the Ministry of Health).

## Verification of CDSUD situations

Precise identification of patients who received CDSUD was essential for comparison purposes. However, an initial analysis revealed differences between physicians' reports and the characteristics of end-of-life care described in the questionnaire responses. Some physicians who reported performing CDSUD provided responses incompatible with this definition, while others, describing care consistent with CDSUD, did not explicitly mention it.

To ensure rigorous identification of CDSUD cases, the questionnaires were analysed in detail by a multidisciplinary team composed of one general practitioner and one palliative physician from among the authors of this study (A.E. and H.B., practicing in a palliative care unit and in a private practice, respectively); and a demographer (S.P.).

They first selected questionnaires where the physicians had explicitly reported making the decision to implement CDSUD (question 25) and/or those where the final medical act was described as CDSUD (question 32).

The two physicians then made an independent analysis of these questionnaires using complementary methodologies. The first applied search equations based on the responses to questions 17, 25, and 32 (S3 File). The second reviewed all responses in each selected questionnaire.

Each questionnaire was independently classified into one of the following five categories: CDSUD certain, highly probable, probable, uncertain, or unlikely.

Differences between the initial classifications of the two physicians were discussed in detail. After comparing their evaluations, the physicians independently re-examined the questionnaires, and this resolved some differences. Cases of persistent disagreement were submitted for collegial analysis, including the demographer, to reach a consensus.

 

Cases were finally classified as confirmed CDSUD if considered certain, highly probable or probable, and classified as no CDSUD if considered uncertain or unlikely (S4 File).

## Analysis of results

The following variables were used to compare patients' characteristics: sex, age, rural or urban residence, overseas department of death, place of death, underlying cause of death, and presence of cognitive impairment at the time of death.

The following variables were used to compare physicians' characteristics: sex, age, medical specialty, practice setting and palliative care training.

The following variables were used to describe the characteristics of CDSUD: treatments received, life-sustaining treatments (hydration, nutrition), collegial procedure, duration, and intention.

Last, we explored the physicians' perception of the end of life and their emotional response to CDSUD.

## Representativeness of responses

To ensure the representativeness of the deaths examined in the study, data were adjusted and standardized based on sex, age, place of death, data collection period, and overseas department.

The differences between the responses and the total number of deaths over the period were minimal in terms of gender, age group and place of death. Unsurprisingly, they were slightly higher for later collection periods, probably because we were interviewing some of the same physicians. The main difference was between the departments, with a higher response rate for Réunion (23%) than for the others (15.4–19.7%). We used weighted frequencies and percentages for our results to ensure better representativeness of our initial population.

## Statistical analysis

Categorical data are presented as unweighted and weighted frequencies and percentages. Proportional comparisons were performed using Chi-squared tests or, when appropriate, Fischer tests, with a 5% significance level. Fisher tests were performed on weighted contingency tables without missing or non-applicable values. Cross-tabulations were performed using R software (version 4.4.3).

## Logistic regression

We conducted a logistic regression to better understand the factors associated with receiving CDSUD. The dependent variable in our analysis was defined as "no CDSUD".

Variables with more than 3% missing values or responses categorized as "don't know" or "not applicable", were grouped in a specific category and included in the analysis. When the percentage of missing values was below 3%, no adjustments were made and the questionnaires with missing values were excluded during the model computations. We altered some variables by grouping items to reduce the number of degrees of freedom (S5 File).

In the regression, we selected the variables that were answered in all cases (with or without CDSUD).

The regressions were performed using R software (packages "srvyr", "gtsummary" and "ggstats")

# Results

## Study population

A total of 12,895 deaths were recorded during the inclusion period. 8,730 questionnaires were sent to physicians who had signed the death certificates and 1,815 completed questionnaires were received, distributed as follows: 1,014 from Réunion, 349 from Guadeloupe, 374 from Martinique, 61 from French Guiana, and 17 missing values. The overall

response rate was 22.9%. A total of 1,375.11 questionnaires concerned deaths for which physicians were able to provide end-of-life information (1,407 unweighted), and 1,029.29 concerned non-sudden deaths (1,082 unweighted). Analysis of the questionnaires revealed that 113.26 (128 unweighted) deaths were preceded by a CDSUD, representing 11.00% of the total (11.83% unweighted) (Fig 1). Only weighted frequencies and percentages are used below.

## Decedents' characteristics

Patients receiving CDSUD were younger than the others (p = 0.0005) and half were aged 40–69 (53.31%). There were fewer patients aged 90+ in the CDSUD group than in the no-CDSUD group (5.02% versus 26.13%; p = 0.0005) (Fig 2).

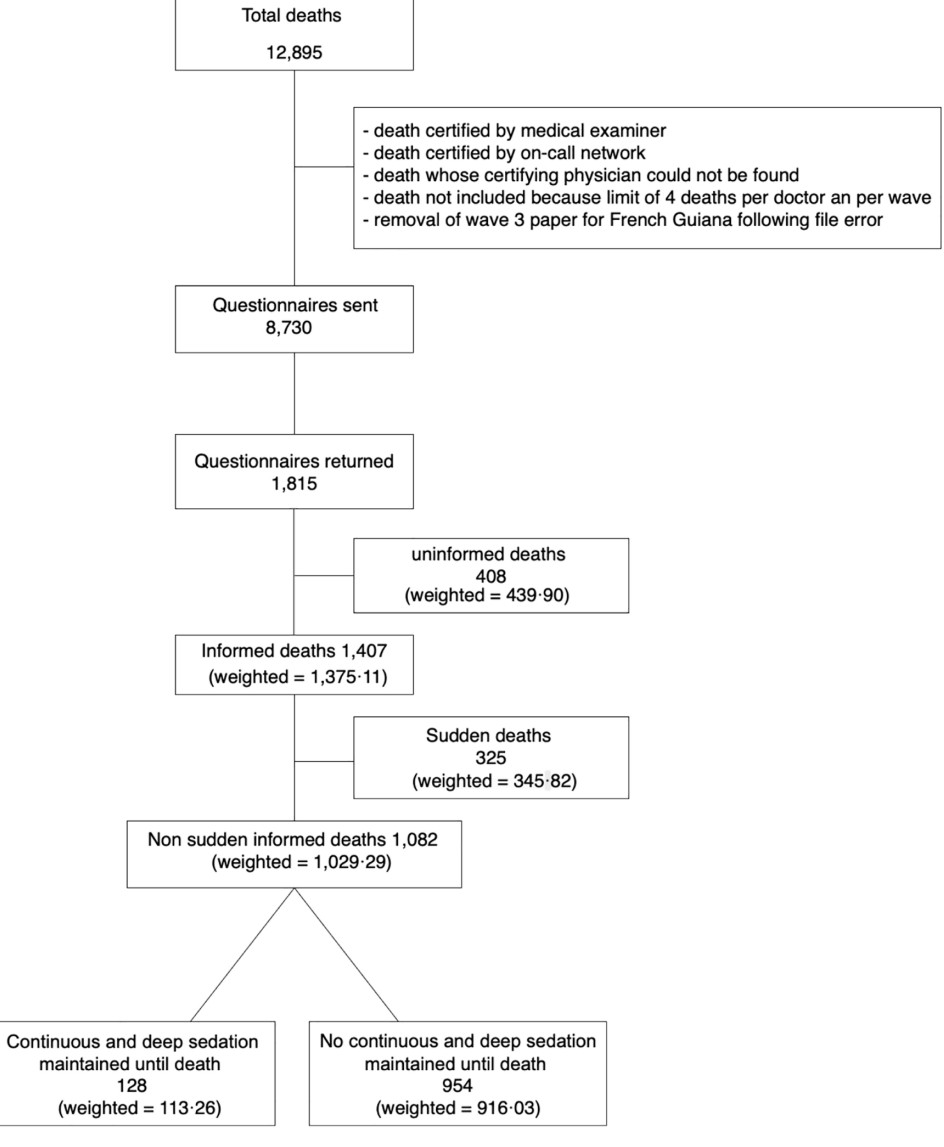

**Fig 1. Flow-chart of data collection and sample.**

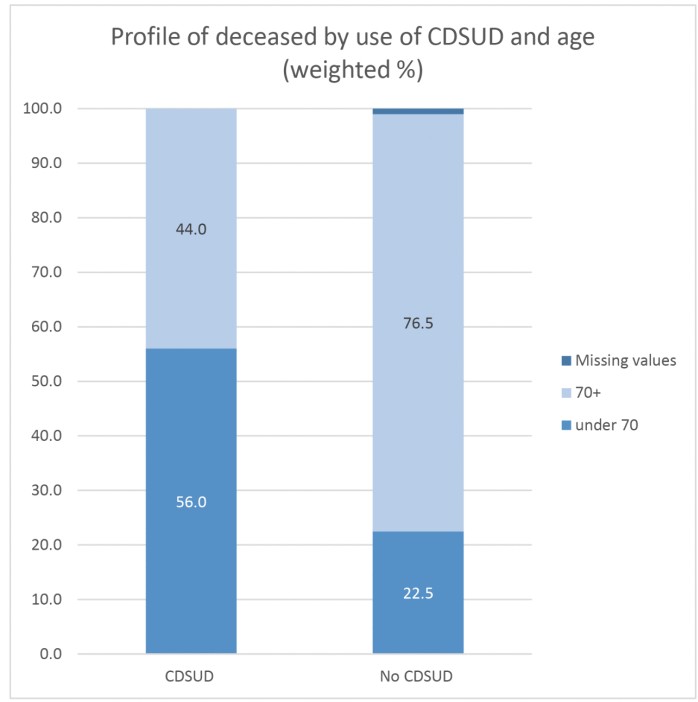 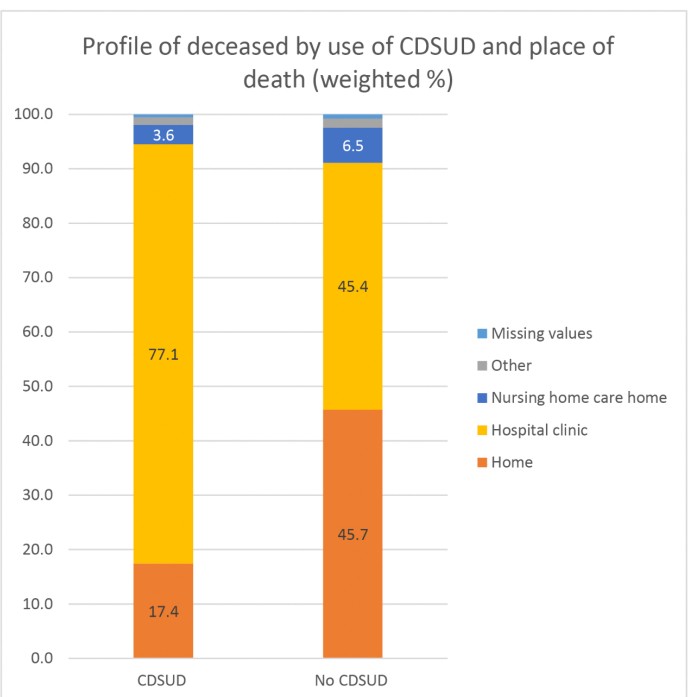

**Fig 2. Age and place of deaths of decedents having received CDSUD.**

CDSUD was performed mainly in Réunion (55.16%; p = 0.0005) and in a hospital or private clinic (77.08%; p = 0.0005). Patients receiving CDSUD were more likely to live in rural areas (56.02% versus 45.73%, p = 0.0430). There was no statistically significant difference in cause of death between the two groups, and the most frequent cause of death was cancer for both groups (34.25% in the CDSUD group versus 33.88%, p = 0.0525). A significantly higher percentage of patients receiving CDSUD had no cognitive impairment (56.45% versus 40.15% in the no-CDSUD group; p = 0.004) (Table 1).

### Physicians' characteristics

Physicians who performed CDSUD were mostly under 40 years of age (43.31%; p = 0.002) and practiced specialty medicine (72.11%; p < 0.0001) other than family medicine (24.95%; p < 0.0001) in a hospital setting (84.46%; p = 0.0005), with no significant differences regarding palliative care training (p = 0.2881) (Table 2).

### CDSUD characteristics

**Treatments received.** Among patients undergoing CDSUD, 78.78% received vigilance-altering treatment with midazolam or another benzodiazepine, and 89.68% received morphine or other opioids.

Artificial hydration was continued until death more frequently in the CDSUD group than in the no-CDSUD group (58.98% versus 48.99%; p = 0.001). However, it was also more frequently discontinued a few hours before death (10.74% versus 4.5%; p = 0.001). Similarly, artificial nutrition was continued more frequently until death in CDSUD group than in the no-CDSUD group (9.98% versus 7.44%; p = 0.001) but was more frequently discontinued a few hours before death (11.94% versus 2.5%; p = 0.001) (Table 3).

**Collegial procedure.** In 79.36% (p < 0.0001) of CDSUD cases, the physician discussed their medical decision with one or more other physicians. Discussions also occurred with the healthcare team in 74.91% (p < 0.0001) of cases, with

**Table 1. Decedents' characteristics.**

| | CDSUD | | | | No CDSUD | | | | Total | | | | Fisher p-value* |
|---|---|---|---|---|---|---|---|---|---|---|---|---|---|
| | N | Column % | weighted N | weighted % | N | % | weighted N | Column weighted % | N | % | weighted N | Column weighted % | |
| Overall | 128 | 100 | 113.26 | 100 | 954 | 100 | 916.03 | 100 | 1082 | 100 | 1029.29 | 100 | |
| **Sex** | | | | | | | | | | | | | 0.0906 |
| Male | 76 | 59.38 | 67.3 | 59.42 | 479 | 50.21 | 463.07 | 50.55 | 555 | 51.29 | 530.37 | 51.53 | |
| Female | 52 | 40.62 | 45.96 | 40.58 | 470 | 49.27 | 448.86 | 49 | 522 | 48.24 | 494.82 | 48.07 | |
| Missing values | 0 | 0 | 0 | 0 | 5 | 0.52 | 4.1 | 0.45 | 5 | 0.46 | 4.1 | 0.4 | |
| **Age (years)** | | | | | | | | | | | | | 0.0005 |
| under 40 | 3 | 2.34 | 3.07 | 2.71 | 22 | 2.31 | 20.66 | 2.26 | 25 | 2.31 | 23.73 | 2.31 | |
| 40-59 | 34 | 26.56 | 28.32 | 25 | 85 | 8.91 | 77.68 | 8.48 | 119 | 11 | 106 | 10.3 | |
| 60-69 | 35 | 27.34 | 32.07 | 28.31 | 126 | 13.21 | 107.74 | 11.76 | 161 | 14.88 | 139.81 | 13.58 | |
| 70-79 | 25 | 19.53 | 21.93 | 19.37 | 195 | 20.44 | 173.97 | 18.99 | 220 | 20.33 | 195.9 | 19.03 | |
| 80-89 | 25 | 19.53 | 22.19 | 19.59 | 293 | 30.71 | 287.42 | 31.38 | 318 | 29.39 | 309.61 | 30.08 | |
| 90+ | 6 | 4.69 | 5.68 | 5.02 | 223 | 23.38 | 239.39 | 26.13 | 229 | 21.16 | 245.07 | 23.81 | |
| Missing values | 0 | 0 | 0 | 0 | 10 | 1.05 | 9.17 | 1 | 10 | 0.92 | 9.17 | 0.89 | |
| **Place of residence** | | | | | | | | | | | | | 0.043 |
| Urban area | 46 | 35.94 | 40.51 | 35.77 | 442 | 46.33 | 450.42 | 49.17 | 488 | 45.1 | 490.93 | 47.7 | |
| Rural area | 73 | 57.03 | 63.45 | 56.02 | 468 | 49.06 | 418.9 | 45.73 | 541 | 50 | 482.35 | 46.86 | |
| Don't know | 5 | 3.91 | 5.41 | 4.78 | 32 | 3.35 | 32.98 | 3.6 | 37 | 3.42 | 38.39 | 3.73 | |
| Missing values | 4 | 3.12 | 3.89 | 3.43 | 12 | 1.26 | 13.73 | 1.5 | 16 | 1.48 | 17.62 | 1.71 | |
| **Department of death** | | | | | | | | | | | | | 0.0005 |
| Guadeloupe | 8 | 6.25 | 11.98 | 10.57 | 148 | 15.51 | 204.08 | 22.28 | 156 | 14.42 | 216.06 | 20.99 | |
| Guyane | 9 | 7.03 | 17.3 | 15.28 | 22 | 2.31 | 42.01 | 4.59 | 31 | 2.87 | 59.31 | 5.76 | |
| Martinique | 16 | 12.5 | 19.23 | 16.98 | 214 | 22.43 | 268.29 | 29.29 | 230 | 21.26 | 287.52 | 27.93 | |
| Réunion | 93 | 72.66 | 62.47 | 55.16 | 563 | 59.01 | 394.64 | 43.07 | 656 | 60.63 | 457.11 | 44.42 | |
| Missing values | 2 | 1.56 | 2.28 | 2.01 | 7 | 0.73 | 7.01 | 0.77 | 9 | 0.83 | 9.29 | 0.9 | |
| **Place of death** | | | | | | | | | | | | | 0.0005 |
| Home | 21 | 16.41 | 19.74 | 17.43 | 420 | 44.03 | 418.92 | 45.73 | 441 | 40.76 | 438.66 | 42.62 | |
| Hospital/clinic | 99 | 77.34 | 87.31 | 77.08 | 434 | 45.49 | 415.7 | 45.38 | 533 | 49.26 | 503.01 | 48.86 | |
| Nursing home/care home | 6 | 4.69 | 4.04 | 3.57 | 82 | 8.6 | 59.18 | 6.46 | 88 | 8.13 | 63.22 | 6.14 | |
| Other | 1 | 0.78 | 1.55 | 1.37 | 11 | 1.15 | 15.08 | 1.65 | 12 | 1.11 | 16.63 | 1.62 | |
| Missing values | 1 | 0.78 | 0.62 | 0.55 | 7 | 0.73 | 7.16 | 0.78 | 8 | 0.74 | 7.78 | 0.76 | |
| **Underlying cause of death** | | | | | | | | | | | | | 0.0525 |
| Cancer | 42 | 32.81 | 38.8 | 34.25 | 328 | 34.38 | 310.36 | 33.88 | 370 | 34.2 | 349.16 | 33.93 | |
| Cardiovascular disease | 13 | 10.16 | 10.45 | 9.23 | 188 | 19.71 | 181.11 | 19.77 | 201 | 18.58 | 191.56 | 18.61 | |
| Neurological or cerebrovascular disease | 30 | 23.44 | 23.69 | 20.92 | 160 | 16.77 | 158.64 | 17.32 | 190 | 17.56 | 182.33 | 17.71 | |
| Infectious disease | 18 | 14.06 | 17.68 | 15.61 | 122 | 12.79 | 114.13 | 12.46 | 140 | 12.94 | 131.81 | 12.81 | |
| Respiratory disease | 12 | 9.38 | 9.49 | 8.38 | 52 | 5.45 | 49.73 | 5.43 | 64 | 5.91 | 59.22 | 5.75 | |
| Digestive disease | 8 | 6.25 | 6.34 | 5.6 | 26 | 2.73 | 25.49 | 2.78 | 34 | 3.14 | 31.83 | 3.09 | |
| Mental/psychiatric disorder | 1 | 0.78 | 1.97 | 1.74 | 24 | 2.52 | 26.85 | 2.93 | 25 | 2.31 | 28.82 | 2.8 | |
| Other causes | 3 | 2.34 | 2.95 | 2.6 | 40 | 4.19 | 34.51 | 3.77 | 43 | 3.97 | 37.46 | 3.64 | |

*(Continued)*

**Table 1.** (Continued)

| | CDSUD | | | | No CDSUD | | | | Total | | | | Fisher p-value* |
|---|---|---|---|---|---|---|---|---|---|---|---|---|---|
| | N | Column % | weighted N | weighted % | N | % | weighted N | Column weighted % | N | % | weighted N | Column weighted % | |
| Missing values | 1 | 0.78 | 1.89 | 1.67 | 14 | 1.47 | 15.21 | 1.66 | 15 | 1.39 | 17.1 | 1.66 | |
| **Cognitive impairment** | | | | | | | | | | | | | **0.004** |
| No | 75 | 58.59 | 63.94 | 56.45 | 395 | 41.4 | 367.81 | 40.15 | 470 | 43.44 | 431.75 | 41.95 | |
| Mild cognitive disorder | 20 | 15.62 | 20.23 | 17.87 | 205 | 21.49 | 197.58 | 21.57 | 225 | 20.79 | 217.81 | 21.16 | |
| Severe cognitive disorder | 27 | 21.09 | 24.29 | 21.45 | 317 | 33.23 | 313.23 | 34.19 | 344 | 31.79 | 337.52 | 32.79 | |
| Don't know | 6 | 4.69 | 4.79 | 4.23 | 30 | 3.14 | 31.47 | 3.44 | 36 | 3.33 | 36.26 | 3.52 | |
| Missing values | 0 | 0 | 0 | 0 | 7 | 0.73 | 5.94 | 0.65 | 7 | 0.65 | 5.94 | 0.58 | |

**\*** Fisher tests are performed on weighted contingency tables without missing values and non-applicable values.

the patient's spouse and/or family in 66.16% (p = 0.0005) of cases, and with the trusted third party in 42.66% (p = 0.0005) of cases.

However, 36.63% (p < 0.0001) of the physicians reported having implemented a collegial procedure as required by law.

**Duration.** The duration of sedation practices was significantly shorter in the CDSUD group. Sedation lasted for a few hours in 45.84% of cases in the CDSUD group, and in 13.8% of cases in the no-CDSUD group (p = 0.0005) (percentage calculated by removing patients who did not receive treatment altering vigilance or consciousness) (Table 3).

**Intention.** In 50.46% of cases in the CDSUD-group, versus 18.45% in the no-CDSUD group (p = 0.0391), sedation was performed knowing it might hasten death. In 54.63% of cases in the CDSUD group, versus 17.21% in the no-CDSUD group, sedation was implemented with the aim of removing the patient's perception of unbearable suffering (p = 0.0015) (Table 3). Very few physicians administered sedation with the deliberate intention to hasten death (2.44% in the CDSUD-group versus 0.17% in the no-CDSUD group; p = 0.0681).

## Physician's perceptions of end-of-life and impact of CDSUD

A significant proportion of physicians did not express an opinion on whether the conditions of death were in line with the patient's expectations (47.92% in the CDSUD group versus 37.72% in the no CDSUD group; p = 0.0935)).

Among those who expressed an opinion, most deaths were in line with patient's expectations, with no statistically significant difference between the two groups (82.51% in the CDSUD-group versus 86.27% in no-CDSUD group; p = 0.0935).

For at least 80% of physicians, the conditions in which end-of-life care took place were rated as "very good" or "fairly good", with no statistically significant difference between the CDSUD group and the others (p = 0.1944).

The emotional impact of patient death on physicians did not differ significantly between the patients who received CDSUD and those who did not (p = 0.4298) (Table 4) (S6 File).

## Discussion

### Main findings

Our study is the first to describe CDSUD within a general population, including deaths at home, irrespective of the cause and place of death. CDSUD was implemented in 11.00% of non-sudden deaths, mainly in hospital settings and in younger patients without cognitive impairment. Morphine was used in 9 cases out of ten and midazolam or another benzodiazepine was used in almost 8 cases out of ten. Collegial procedures were not systematic, although physicians discussed

**Table 2. Physicians' characteristics.**

| | CDSUD | | | | No CDSUD | | | | Total | | | | Fisher p-value* |
|---|---|---|---|---|---|---|---|---|---|---|---|---|---|
| | N | Column % | weighted N | Column weighted % | N | Column % | weighted N | Column weighted % | N | Column % | weighted N | Column weighted % | |
| Overall | 128 | 100 | 113.26 | 100 | 954 | 100 | 916.03 | 100 | 1082 | 100 | 1029.29 | 100 | |
| Sex | | | | | | | | | | | | | 0.8399 |
| Male | 61 | 47.66 | 49.96 | 44.11 | 413 | 43.29 | 398.82 | 43.54 | 474 | 43.81 | 448.78 | 43.6 | |
| Female | 66 | 51.56 | 61.35 | 54.17 | 534 | 55.97 | 510.28 | 55.71 | 600 | 55.45 | 571.63 | 55.54 | |
| Missing values | 1 | 0.78 | 1.95 | 1.72 | 7 | 0.73 | 6.93 | 0.76 | 8 | 0.74 | 8.88 | 0.86 | |
| Age (years) | | | | | | | | | | | | | 0.002 |
| Under 40 | 52 | 40.62 | 49.05 | 43.31 | 316 | 33.12 | 319.36 | 34.86 | 368 | 34.01 | 368.41 | 35.79 | |
| 40-49 | 37 | 28.91 | 30.57 | 27 | 200 | 20.96 | 180.26 | 19.68 | 237 | 21.9 | 210.83 | 20.48 | |
| 50-59 | 30 | 23.44 | 24.34 | 21.5 | 214 | 22.43 | 208.49 | 22.76 | 244 | 22.55 | 232.83 | 22.62 | |
| 60+ | 9 | 7.03 | 9.28 | 8.2 | 216 | 22.64 | 200.73 | 21.91 | 225 | 20.79 | 210.01 | 20.4 | |
| Missing values | 0 | 0 | 0 | 0 | 8 | 0.84 | 7.21 | 0.79 | 8 | 0.74 | 7.21 | 0.7 | |
| Medical Specialty | | | | | | | | | | | | | <0.0001 |
| General practitioner | 33 | 25.78 | 28.26 | 24.95 | 602 | 63.1 | 587.12 | 64.09 | 635 | 58.69 | 615.38 | 59.79 | |
| Other specialty | 92 | 71.88 | 81.67 | 72.11 | 336 | 35.22 | 314.79 | 34.36 | 428 | 39.56 | 396.46 | 38.52 | |
| Missing values | 3 | 2.34 | 3.33 | 2.94 | 16 | 1.68 | 14.12 | 1.54 | 19 | 1.76 | 17.45 | 1.7 | |
| Place of practice | | | | | | | | | | | | | 0.0005 |
| Independent practice | 19 | 14.84 | 16.97 | 14.98 | 374 | 39.2 | 357.59 | 39.04 | 393 | 36.32 | 374.56 | 36.39 | |
| Hospital | 108 | 84.38 | 95.66 | 84.46 | 526 | 55.14 | 499.76 | 54.56 | 634 | 58.6 | 595.42 | 57.85 | |
| Mixed structure | 0 | 0 | 0 | 0 | 3 | 0.31 | 5.04 | 0.55 | 3 | 0.28 | 5.04 | 0.49 | |
| Independent and hospital | 0 | 0 | 0 | 0 | 8 | 0.84 | 9.6 | 1.05 | 8 | 0.74 | 9.6 | 0.93 | |
| Home care | 0 | 0 | 0 | 0 | 35 | 3.67 | 35.9 | 3.92 | 35 | 3.23 | 35.9 | 3.49 | |
| Other | 1 | 0.78 | 0.63 | 0.55 | 8 | 0.84 | 8.14 | 0.89 | 9 | 0.83 | 8.77 | 0.85 | |
| Palliative care training | | | | | | | | | | | | | 0.2881 |
| Yes, graduate training | 38 | 29.69 | 31.77 | 28.05 | 221 | 23.17 | 220.6 | 24.08 | 259 | 23.94 | 252.37 | 24.52 | |
| Yes, in service training | 16 | 12.5 | 12.51 | 11.04 | 168 | 17.61 | 157.59 | 17.2 | 184 | 17.01 | 170.1 | 16.53 | |
| No | 73 | 57.03 | 1.89 | 1.67 | 541 | 56.71 | 10.1 | 1.1 | 614 | 56.75 | 11.99 | 1.16 | |
| Both graduate and in-service | 1 | 0.78 | 67.09 | 59.24 | 11 | 1.15 | 515.23 | 56.25 | 12 | 1.11 | 582.32 | 56.57 | |
| Missing values | 0 | 0 | 0 | 0 | 13 | 1.36 | 12.51 | 1.37 | 13 | 1.2 | 12.51 | 1.22 | |

\* Fisher tests are performed on weighted contingency tables without missing values and non-applicable values.

their decisions with a colleague in 80% of cases. CDSUD was mainly performed with the knowledge that it might hasten death, without the deliberate intention to cause death and with the aim of removing the patient's perception of unbearable suffering. There was no difference in the emotional impact of the death on the physician, whether or not the patient had received CDSUD.

## What this study adds

**CDSUD prevalence.** Our study provides data on prevalence of CDSUD in the specific geographical context of the French overseas departments, geographical areas that share the same organizational environment as mainland France, but also have specific insular features such as a more rapidly ageing population, very few nursing homes and therefore more home deaths. These departments also have higher levels of religiosity. These particularities tend to favour a lower

**Table 3. Characteristics of practices involving alteration of consciousness.**

| | CDSUD | | | | No CDSUD | | | | Total | | | | Fisher p-value* |
|---|---|---|---|---|---|---|---|---|---|---|---|---|---|
| | N | Column % | weighted N | Column weighted % | N | Column % | weighted N | Column weighted % | N | Column % | weighted N | Column weighted % | |
| Overall | 128 | 100 | 113.26 | 100 | 954 | 100 | 916.03 | 100 | 1082 | 100 | 1029.29 | 100 | |
| **Received a treatment altering vigilance or consciousness** | | | | | | | | | | | | | <0.0001 |
| Yes | 124 | 96.88 | 108.68 | 95.96 | 435 | 45.6 | 400.36 | 43.71 | 559 | 51.66 | 509.04 | 49.45 | |
| No | 4 | 3.12 | 4.58 | 4.04 | 505 | 52.94 | 503.27 | 54.94 | 509 | 47.04 | 507.85 | 49.34 | |
| Don't know | 0 | 0 | 0 | 0 | 7 | 0.73 | 6.93 | 0.76 | 7 | 0.65 | 6.93 | 0.67 | |
| Missing values | 0 | 0 | 0 | 0 | 7 | 0.73 | 5.47 | 0.6 | 7 | 0.65 | 5.47 | 0.53 | |
| **Drugs used to alter vigilance or consciousness (multiple responses)** | | | | | | | | | | | | | |
| Morphine or other opioids | | 89.84 | 101.57 | 89.68 | 389 | 40.78 | 361.98 | 39.52 | 504 | 46.58 | 463.55 | 45.04 | 0.4457 |
| Midazolam or other benzodiazepines | 104 | 81.25 | 89.22 | 78.78 | 197 | 20.65 | 175.59 | 19.17 | 301 | 27.82 | 264.81 | 25.73 | <0.0001 |
| Other hypnotics | 7 | 5.47 | 4.7 | 4.15 | 2 | 0.21 | 1.24 | 0.14 | 9 | 0.83 | 5.94 | 0.58 | 0.0021 |
| Antidepressant | 0 | 0 | 0 | 0 | 1 | 0.1 | 0.63 | 0.07 | 1 | 0.09 | 0.63 | 0.06 | 1 |
| Antipsychotic | 2 | 1.56 | 2.43 | 2.14 | 4 | 0.42 | 3.84 | 0.42 | 6 | 0.55 | 6.27 | 0.61 | 0.6119 |
| Anticholinergic | 1 | 0.78 | 0.67 | 0.59 | 8 | 0.84 | 7.54 | 0.82 | 9 | 0.83 | 8.21 | 0.8 | 0.6917 |
| Other | 0 | 0 | 0 | 0 | 2 | 0.21 | 1.44 | 0.16 | 2 | 0.18 | 1.44 | 0.14 | 1 |
| **Duration of alteration of vigilance or consciousness** | | | | | | | | | | | | | 0.0005 |
| A few hours | 59 | 46.09 | 51.92 | 45.84 | 70 | 7.34 | 55.29 | 6.04 | 129 | 11.92 | 107.21 | 10.42 | |
| A few days | 40 | 31.25 | 32.14 | 28.37 | 212 | 22.22 | 194.82 | 21.27 | 252 | 23.29 | 226.96 | 22.05 | |
| A few weeks | 21 | 16.41 | 20.62 | 18.21 | 144 | 15.09 | 142.88 | 15.6 | 165 | 15.25 | 163.5 | 15.88 | |
| Missing values | 4 | 3.12 | 4 | 3.54 | 9 | 0.94 | 7.36 | 0.8 | 13 | 1.2 | 11.36 | 1.1 | |
| No to question on alteration of vigilance or consciousness | 4 | 3.12 | 4.58 | 4.04 | 519 | 54.4 | 515.68 | 56.29 | 523 | 48.34 | 520.26 | 50.55 | |
| **Intentionality (multiple responses)** | | | | | | | | | | | | | |
| Knowing it won't hasten death: Yes | 22 | 17.19 | 22.74 | 20.08 | 96 | 10.06 | 93.66 | 10.22 | 118 | 10.91 | 116.4 | 11.31 | 0.7002 |
| Knowing it might hasten death: Yes | 64 | 50 | 57.15 | 50.46 | 191 | 20.02 | 164.8 | 17.99 | 255 | 23.57 | 221.95 | 21.56 | 0.0391 |
| No concern it might hasten death: Yes | 41 | 32.03 | 34.75 | 30.68 | 111 | 11.64 | 112.42 | 12.27 | 152 | 14.05 | 147.17 | 14.3 | 0.406 |
| With express intention to hasten death: Yes | 4 | 3.12 | 2.77 | 2.44 | 2 | 0.21 | 1.58 | 0.17 | 6 | 0.55 | 4.35 | 0.42 | 0.0681 |
| With intention to remove the patient's perception of unbearable suffering: Yes | 72 | 56.25 | 61.87 | 54.63 | 175 | 18.34 | 157.64 | 17.21 | 247 | 22.83 | 219.51 | 21.33 | 0.0015 |
| With another intention: yes | 0 | 0 | 0 | 0 | 1 | 0.1 | 1.51 | 0.16 | 1 | 0.09 | 1.51 | 0.15 | 1 |

*(Continued)*

**Table 3.** (Continued)

| | CDSUD | | | | No CDSUD | | | | Total | | | | Fisher p-value* |
|---|---|---|---|---|---|---|---|---|---|---|---|---|---|
| | N | Column % | weighted N | Column weighted % | N | Column % | weighted N | Column weighted % | N | Column % | weighted N | Column weighted % | |
| *Life sustaining treatment* | | | | | | | | | | | | | |
| **Artificial hydration** | | | | | | | | | | | | | 0.001 |
| Yes, continuously until death | 81 | 63.28 | 66.8 | 58.98 | 472 | 49.48 | 448.77 | 48.99 | 553 | 51.11 | 515.57 | 50.09 | |
| Yes, but discontinued a few hours before death | 12 | 9.38 | 12.17 | 10.74 | 49 | 5.14 | 41.24 | 4.5 | 61 | 5.64 | 53.41 | 5.19 | |
| Yes, but discontinued a few days before death | 8 | 6.25 | 8.95 | 7.9 | 74 | 7.76 | 67.86 | 7.41 | 82 | 7.58 | 76.81 | 7.46 | |
| Yes, but discontinued a few weeks before death | 4 | 3.12 | 4.58 | 4.04 | 18 | 1.89 | 20.36 | 2.22 | 22 | 2.03 | 24.94 | 2.42 | |
| No | 23 | 17.97 | 20.76 | 18.34 | 322 | 33.75 | 318.38 | 34.76 | 345 | 31.89 | 339.14 | 32.95 | |
| Don't know | 0 | 0 | 0 | 0 | 17 | 1.78 | 17.86 | 1.95 | 17 | 1.57 | 17.86 | 1.74 | |
| Missing values | 0 | 0 | 0 | 0 | 2 | 0.21 | 1.56 | 0.17 | 2 | 0.18 | 1.56 | 0.15 | |
| **Artificial Nutrition** | | | | | | | | | | | | | 0.001 |
| Yes, continuously until death | 15 | 11.72 | 11.3 | 9.98 | 72 | 7.55 | 68.16 | 7.44 | 87 | 8.04 | 79.46 | 7.72 | |
| Yes, but discontinued a few hours before death | 18 | 14.06 | 13.53 | 11.94 | 28 | 2.94 | 22.91 | 2.5 | 46 | 4.25 | 36.44 | 3.54 | |
| Yes, but discontinued a few days before death | 7 | 5.47 | 5.97 | 5.27 | 47 | 4.93 | 45.35 | 4.95 | 54 | 4.99 | 51.32 | 4.99 | |
| Yes, but discontinued a few weeks before death | 4 | 3.12 | 5.03 | 4.45 | 11 | 1.15 | 11.17 | 1.22 | 15 | 1.39 | 16.2 | 1.57 | |
| No | 82 | 64.06 | 74.8 | 66.04 | 779 | 81.66 | 751.96 | 82.09 | 861 | 79.57 | 826.76 | 80.32 | |
| Don't know | 2 | 1.56 | 2.63 | 2.32 | 15 | 1.57 | 14.92 | 1.63 | 17 | 1.57 | 17.55 | 1.71 | |
| Missing values | 0 | 0 | 0 | 0 | 2 | 0.21 | 1.56 | 0.17 | 2 | 0.18 | 1.56 | 0.15 | |

*Fisher tests are performed on weighted contingency tables without missing values and non-applicable values.

level of technical medical decisions, such as CDSUD. Recent legal changes have nonetheless been implemented in these departments, as about one in eight non-sudden deaths were preceded by CDSUD. Although the prevalence rates differ significantly among the overseas departments, the overall number of cases warrants caution in interpreting these findings as indicative of differences in clinical practice.

The existing literature concerning this practice remains limited, with reported prevalence in France varying considerably (from 0.5% for overall deaths to 60% in intensive care units after a decision of withdraw life sustaining therapies) depending on the populations studied. [7,8,10,11] This heterogeneity is also observed in Europe (2.5% to 8.5%), mainly due to variations in study populations, legislation and definitions of CDSUD. [19–21] These data are open to criticism, as they may be distorted by reporting bias, but a double reading of the questionnaires nevertheless seems to confirm a high prevalence.

**Decedents' characteristics.** Patients who received CDSUD were significantly younger and had less cognitive impairment than those who did not. The prevalence of CDSUD was significantly lower among individuals aged 90 + . The young age of patients receiving CDSUD is widely reported in the existing literature. [5,19,22] In line with national mortality statistics in France, cancer was the most frequent cause of death in both groups, suggesting that CDSUD is implemented

**Table 4. Physician's perceptions of end-of-life and impact of sedative practices.**

| | CDSUD | | | | No CDSUD | | | | Total | | | | Fisher p-value* |
|---|---|---|---|---|---|---|---|---|---|---|---|---|---|
| | N | Column % | weighted N | Column weighted % | N | Column % | weighted N | Column weighted % | N | Column % | weighted N | Column weighted % | |
| Overall | 128 | 100 | 113.26 | 100 | 954 | 100 | 916.03 | 100 | 1082 | 100 | 1029.29 | 100 | |
| **Deaths in line with patients' expectations** | | | | | | | | | | | | | 0.0935 |
| Yes | 60 | 46.88 | 48.67 | 42.97 | 517 | 54.19 | 494.59 | 53.99 | 577 | 53.33 | 543.26 | 52.78 | |
| No | 8 | 6.25 | 7.14 | 6.31 | 55 | 5.77 | 54.51 | 5.95 | 63 | 5.82 | 61.65 | 5.99 | |
| Don't know | 57 | 44.53 | 54.27 | 47.92 | 356 | 37.32 | 345.51 | 37.72 | 413 | 38.17 | 399.78 | 38.84 | |
| Missing values | 3 | 2.34 | 3.18 | 2.81 | 26 | 2.73 | 21.42 | 2.34 | 29 | 2.68 | 24.6 | 2.39 | |
| **End-of-life care conditions according to physicians** | | | | | | | | | | | | | 0.1944 |
| Very good | 38 | 29.69 | 31.73 | 28.01 | 288 | 30.19 | 282.93 | 30.89 | 326 | 30.13 | 314.66 | 30.57 | |
| Fairly good | 77 | 60.16 | 71.39 | 63.03 | 507 | 53.14 | 479.92 | 52.39 | 584 | 53.97 | 551.31 | 53.56 | |
| Poor | 7 | 5.47 | 6.2 | 5.47 | 55 | 5.77 | 56.24 | 6.14 | 62 | 5.73 | 62.44 | 6.07 | |
| Very poor | 1 | 0.78 | 0.61 | 0.54 | 15 | 1.57 | 14.82 | 1.62 | 16 | 1.48 | 15.43 | 1.5 | |
| Missing values | 5 | 3.91 | 3.33 | 2.94 | 89 | 9.33 | 82.12 | 8.97 | 94 | 8.69 | 85.45 | 8.3 | |
| **Emotional impact of patient death for physician (scale one to ten)** | | | | | | | | | | | | | 0.4298 |
| 0 | 23 | 17.97 | 21.49 | 18.98 | 130 | 13.63 | 135.5 | 14.79 | 153 | 14.14 | 156.99 | 15.25 | |
| 1-3 | 49 | 38.28 | 39.41 | 34.8 | 408 | 42.77 | 387.28 | 42.28 | 457 | 42.24 | 426.69 | 41.45 | |
| 4-5 | 29 | 22.66 | 28.01 | 24.73 | 209 | 21.91 | 196.12 | 21.41 | 238 | 22 | 224.13 | 21.78 | |
| 5+ | 26 | 20.31 | 23.72 | 20.94 | 181 | 18.97 | 169.84 | 18.54 | 207 | 19.13 | 193.56 | 18.81 | |
| Missing values | 1 | 0.78 | 0.62 | 0.55 | 26 | 2.73 | 27.3 | 2.98 | 27 | 2.5 | 27.92 | 2.71 | |

* Fisher tests are performed on weighted contingency tables without missing values and non-applicable values.

in response to refractory symptoms rather than a specific pathology. [15] Cognitive impairment was associated with a lower prevalence of CDSUD, probably because of difficulties obtaining informed consent in this population, and estimating their suffering, as found in previous studies concerning pain. [23]

In three-quarters of cases, CDSUD took place in a hospital setting. This may be linked to the complexity of its implementation at home, for which a 24/7 care team is required. [24] Most older people die at home, and this may also explain the under-representation of this population among patients receiving CDSUD.

**Physicians' characteristics.** Physicians who performed CDSUD were significantly younger than those who did not and practiced a medical specialty in a hospital setting. There are few data in the literature on physicians performing CDSUD. A study by Pennec et al. in 2005 found that physicians under 40 years old were more likely to intensify the alleviation of symptoms at the end of life and more likely to withdraw treatments. [13] This younger age may reflect a better integration of the recent legal framework on CDSUD into the training of newer generations of physicians. The absence of any significant difference in palliative care training between the two groups of physicians is a surprising finding. This may suggest that other factors, such as clinical experience or mentorship, play a more decisive role than formal training. Further studies are required to draw any conclusions.

**CDSUD characteristics.** Most patients who received CDSUD were given midazolam or another benzodiazepine and an analgesic treatment with morphine or another opioid, in accordance with guidelines. [1,4]

Guidelines indicate that during CDSUD, all life-sustaining treatments must be withdrawn to avoid prolonged sedation. [1,4] In our study, artificial hydration and nutrition were not consistently withdrawn, unlike in another French intensive care study where withdrawal was systematic. [7] Several studies highlight the difficulty for physicians of withdrawing these treatments, particularly when relatives fear a decline in the patient's quality of life and worry that they could die of thirst. [25]

In most cases, physicians discussed their decision to implement CDSUD with at least one other physician but only about a third followed a collegial procedure in accordance with the law. Some physicians may consider that an informal discussion is sufficient to rule out CDSUD indications. [2]

Most patients receiving CDSUD were sedated for a few hours, consistent with the literature which reports an average duration of sedation of between 48 and 72 hours. [6,7,10,22]

The duration of sedation was significantly shorter in the CDSUD group, possibly due to the more frequent withdrawal of life-sustaining treatment, or an indication bias whereby CDSUD must only be implemented in patients with a short-term prognosis of death.

The intention of CDSUD is an ethically crucial aspect. [26] In about half of cases, the alteration of consciousness was performed with the knowledge that it might potentially hasten the occurrence of death, and in a slightly larger proportion, with the aim of removing the perception of unbearable suffering, illustrating the principle of double effect. The explicit intention to hasten the occurrence of death was very rare in both groups, underlining the clear distinction between CDSUD and euthanasia in physicians' minds.

Contrary to concerns often expressed regarding CDSUD, the perception of end-of-life conditions and the emotional impact of sedative practices on physicians did not differ significantly between the CDSUD and no-CDSUD groups. [27] This suggests that CDSUD, when perceived as an appropriate response to refractory suffering, does not generate an additional emotional burden for physicians. As sedation is mainly carried out in hospital, with round-the-clock monitoring, the burden is shared and therefore may be less stressful than if carried out at home. However, the high non-response rate to the question about the alignment with patient expectations may reflect a lack of understanding of patients' end-of-life expectations and the sensitivity of the topic. Qualitative research could explore in greater depth the emotional experiences of caregivers facing CDSUD and the factors associated with a more serene experience of this practice. [28]

## Strengths and limitations

The multidisciplinary approach combining medical and socio-demographic expertise is a major strength of our study. Furthermore, the rigorous multi-step analysis method used to confirm CDSUD cases allows us to go beyond initial physician reports and significantly reduces the risk of classification bias. The robustness of our statistical method ensured complete and reliable data anonymity, and adjustment for response rates enhances representativeness. Our study presents the largest population-based sample in research on CDSUD.

However, our study also has limitations. We did not have variables on clinical details such as disease stage, comorbidities or symptom burden, and this may have limited our ability to determine the appropriateness of CDSUD. The interpretation of data concerning the collegial procedure and intention may have been altered by the a posteriori reclassification of certain end-of-life situations: physicians who did not initially identify their practice as CDSUD might not have reported a collegial procedure or expressed an intention corresponding to CDSUD. The recall bias inherent to the retrospective nature of the study is another limitation, despite our efforts to minimize it. The timing of the study, partially overlapping with the onset of the COVID-19 pandemic, might have influenced practices, although the extent of this impact remains uncertain, and the low overall response rate could introduce selection bias. While differences in prevalence between territories may exist, the limited number of cases precluded a territory-specific analysis.

Finally, although the legal framework of overseas France is identical to that of mainland France, specific cultural features and modes of healthcare organization, notably illustrated by the higher proportion of home deaths in overseas

France, might limit the external validity of our results. [16] The more rapid demographic ageing in the French overseas departments offers interesting perspectives for anticipating future trends in mainland France. [29]

## Conclusion

Following the legal recognition of CDSUD in France, this study reveals a notable prevalence of CDSUD in the French overseas departments, regions that share the same legal framework as mainland France but with distinct demographic and health characteristics, and confirms its integration into end-of-life care in these departments. Patients receiving CDSUD were more likely to be younger and without cognitive impairment than those who did not, and the practice predominantly occurred in a hospital setting. These results highlight the need to improve access to CDSUD for the elderly and for in-home care. While the use of recommended medications such as midazolam and opioids was common, the systematic withdrawal of artificial hydration and nutrition remains an area for improvement. Discussion of the decision was frequent, although not always organized as a formal collegial procedure, indicating another area where further progress is needed. Importantly, while physicians acknowledged the potential for CDSUD to hasten death in a significant proportion of cases, the primary intention was overwhelmingly to alleviate unbearable suffering, underscoring the ethical distinction with euthanasia. Training and support, particularly from teams specializing in palliative care, could potentially improve the formalization of these collegial procedures. The perception of end-of-life care and the emotional impact on physicians did not significantly differ between cases with and without CDSUD. Future research should focus on prospective data collection and in-depth qualitative studies exploring physician experiences, patient and family perspectives, and institutional factors influencing CDSUD practices. Comparative studies between mainland and overseas regions could shed light on the contextual factors shaping CDSUD integration. Additionally, longitudinal investigations on how CDSUD impacts the quality of end-of-life care and bereavement experiences could further inform best practices and public health policies

## Supporting information

**S1 File. Explanatory booklet presenting the study.**
(DOCX)

**S2 File. Questionnaire sent to physicians.**
(DOCX)

**S3 File. Search equations.**
(DOCX)

**S4 File. Physicians' ranking of questionnaires.**
(DOCX)

**S5 File. Logistic regression.**
(DOCX)

**S6 File. Physician's perceptions of end-of-life and impact of sedative practices.**
(DOCX)

## Acknowledgments

We want to thank all the persons involved in the data collection at Ined, Regional Health Agency (ARS) La Réunion, The hospital of Besançon (CHRU), the center for epidemiology of medical causes of death (CepiDc) of the French institute of health and medical research; the physicians from La Réunion whom we discuss some of the results and last but not least

all the physicians who took from their time to answer our questionnaires. Our thanks also to Catriona Dutreuilh for providing professional English language editing.

## Author contributions

**Conceptualization:** Sophie Pennec, Hélèna Briand.

**Data curation:** Sophie Pennec.

**Formal analysis:** Sophie Pennec, Adrien Evin.

**Funding acquisition:** Sophie Pennec.

**Methodology:** Sophie Pennec, Hélèna Briand, Vincent Guion, Adrien Evin.

**Project administration:** Sophie Pennec.

**Supervision:** Sophie Pennec.

**Writing – original draft:** Hélèna Briand.

**Writing – review & editing:** Sophie Pennec, Hélèna Briand, Vincent Guion, Adrien Evin.

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
