## [Decision Letter · Decision Letter 0]

7 Oct 2025

PONE-D-25-49331How is deep and continuous sedation until death practiced in the French overseas departments?PLOS ONE?

Dear Dr. Pennec,

We look forward to receiving your revised manuscript.

Kind regards,

Stefaan Six, Ph.D.

Academic Editor

PLOS ONE

For additional information about PLOS ONE ethical requirements for human subjects research, please refer to http://journals.plos.org/plosone/s/submission-guidelines#loc-human-subjects-research .

 [SP received grants to support the research by the Caisse nationale de solidarité pour l’autonomie (CNSA), as part of the call for projects launched by IReSP (project IReSP-17-Hand8-16) ; by the Fondation de France as part of the call for project “Soigner, soulager, accompagner (2017)”. The Institut national d’études démographiques (INED) was involved in the data collection through its Survey department.

https://iresp.net/

https://www.fondationdefrance.org/fr/

https://www.ined.fr

No funding bodies was involved in the contents of this paper.]. 

4. In the online submission form, you indicated that [The survey dataset analysed in the current study is not publicly available yet due to its embargo. Data can be accessible under reasonable request upon the corresponding author after approval of a proposal and with a signed access agreement.].

Additional Editor Comments (if provided):

Reviewers' comments:

Reviewer's Responses to Questions

**Comments to the Author**

1. Is the manuscript technically sound, and do the data support the conclusions?

Reviewer #1: Yes

Reviewer #2: Yes

2. Has the statistical analysis been performed appropriately and rigorously?

Reviewer #1: Yes

Reviewer #2: Yes

3. Have the authors made all data underlying the findings in their manuscript fully available?

Reviewer #1: Yes

Reviewer #2: Yes

4. Is the manuscript presented in an intelligible fashion and written in standard English?

Reviewer #1: No

Reviewer #2: Yes

Reviewer #1: This study addresses continuous deep sedation until death (CDSUD) as practiced in the French overseas departments. The methodology consisted of surveying physicians who had completed the death certificates and comparing those who performed CDSUD with those who did not. The article is well written, clear, and concise. I have only two comments:

The tables presenting only the weighted frequencies and weighted percentages are not straightforward to interpret. It would be more informative for the reader if a table with three columns were provided: the first including the unweighted counts and percentages (n (%)), the second the weighted frequencies and weighted percentages (n (%)), and the third the p-value.

Considering the importance of the topic and the data collected in the overseas territories, it is regrettable that practitioners working in these territories were not involved in this publication. I am not convinced that physicians (VG and AE) from Nantes (mainland France), authors of this article, would have appreciated researchers from the overseas departments analyzing and using data from their hospital without being associated with the research. It even appears that physicians from La Réunion were approached to discuss the results. There are established teams and palliative care units in these territories, as well as (future) academics seeking to specialize in palliative care; excluding them from the study borders on disregard.

Reviewer #2: Overall assessment

This manuscript offers a valuable and original contribution by investigating continuous deep sedation until death (CDSUD) in the French overseas departments. The study addresses an important knowledge gap and provides new insights into the implementation of a unique legal framework. The design is thoughtful, and the results are of international interest.

That said, certain aspects require clarification or expansion before the paper can reach its full potential. These issues are not fundamental flaws, but they do limit the current strength of the conclusions.

Major comments

Response rate

The relatively low response rate (22.9%) is a key limitation. While this is common in physician surveys, some additional detail on representativeness (e.g., how respondents compared with the overall physician population in the overseas departments) would reassure readers about the reliability of prevalence estimates.

Case classification

The adjudication process for CDSUD cases is innovative, but it introduces subjectivity. A clearer description of how disagreements were handled, or a basic measure of agreement between reviewers, would strengthen the credibility of this approach.

Clinical information

Because the study relies on death certificates, clinical details such as disease stage, comorbidities, and symptom burden are missing. Explicit acknowledgment of this limitation is needed, as it affects interpretation of appropriateness of CDSUD.

Collegial procedure

The finding that only about one-third of cases followed the formal collegial procedure is highly relevant. The discussion would benefit from exploring possible reasons for this gap (e.g., time constraints, institutional barriers, cultural norms) and its implications for policy and practice.

Cultural context

The overseas departments have distinctive cultural and religious environments, which likely influence end-of-life practices. A more developed reflection on this dimension would enrich the discussion and make the findings more interpretable for international readers.

Minor comments

Be consistent in distinguishing weighted vs. unweighted data throughout the text and tables.

Consider adding a simple figure (e.g., prevalence of CDSUD by age or place of death) to complement the detailed tables.

Define French-specific terms (e.g., collegial procedure) to ensure accessibility for international readers.

The introduction and discussion could emphasize more clearly how this study builds upon and differs from previous European and international studies of sedation.

**Do you want your identity to be public for this peer review?** For information about this choice, including consent withdrawal, please see our Privacy Policy

Reviewer #1: **Yes: ** Denis Boucaud-Maitre

Reviewer #2: No

---

## [Author Response · Author response to Decision Letter 1]

2 Nov 2025

Dear Editor and reviewers,

We thank you for your thorough reading of our paper and your suggestions that helped us to improve our manuscript.

You will find below, all our responses to your questions and suggestions.

Followed your request about the role of Funders, can you, please, change the online mention to: “The funders had no role in the study design, data collection and analysis, decision to publish, or preparation of the manuscript”.

We checked the style requirements and hope not having missed any.

Yours sincerely,

The authors

Editor

We have followed the PLOS ONE’s style requirement, Compared to the previous version:

- titles has been transformed in sentence case and only the first letter in capital

- captions, in-text citations of figures, tables, supplementary material has been checked

- in text citations changed to square brackets

- affiliations checked

For your information, the project went through a double layer of ethics. One is The Comité d’Expertise pour les Recherches, les Études et les Évaluations dans le domaine de la Santé [National Expert Committee for Research, Studies and Evaluations in the Field of Health], and the second is the Commission nationale de l’informatique et des libertés [CNIL-National committee on digital and liberty – liberty act.

On one of the Cnil flyer, research on dead persons is mentioned as followed (our translation): “With regard to the processing of data relating to deceased persons, provided that the professional participating in the research is aware of the vital status of the person concerned and that the latter did not object to this in writing during their lifetime, personal data relating to them may be processed for research purposes”. (Methodology reference MR004)

We altered the sentence both in the methods/ethics statement section and in the declaration /ethical standards section to make clearer that the need for consent was waived by a national ethics committee.

As follows:

The Comité d’Expertise pour les Recherches, les Études et les Évaluations dans le domaine de la Santé [National Expert Committee for Research, Studies and Evaluations in the Field of Health], approved the methodology used, waived the need for consent and waived the requirement of medical confidentiality for research purposes (approval in March 2018)

[SP received grants to support the research by the Caisse nationale de solidarité pour l’autonomie (CNSA), as part of the call for projects launched by IReSP (project IReSP-17-Hand8-16) ; by the Fondation de France as part of the call for project “Soigner, soulager, accompagner (2017)”. The Institut national d’études démographiques (INED) was involved in the data collection through its Survey department.

https://iresp.net/

https://www.fondationdefrance.org/fr/

https://www.ined.fr

No funding bodies was involved in the contents of this paper.].

We have included this statement on the online financial statement box

We have included this statement on the cover letter

4. In the online submission form, you indicated that [The survey dataset analysed in the current study is not publicly available yet due to its embargo. Data can be accessible under reasonable request upon the corresponding author after approval of a proposal and with a signed access agreement.].

We understand your point and Ined as a research institute and data provider is sharing the principle of “open science” and open data as much as possible. In that particular case, some restrictions applies for legal reasons.

We altered the online submission form with the following statement:

Data cannot be shared freely because the department of residence/death can be identified from our dataset and that is considered identifying information by the French liberty act (CNIL). Data can be accessed by researchers affiliated to a research/educational institution, upon submission of a research project through the data progedo plateform (https://data.progedo.fr/). The European General Data Protection Regulation (GDPR) applies.

Additional Editor Comments (if provided):

Reviewers' comments:

Reviewer's Responses to Questions

Comments to the Author

1. Is the manuscript technically sound, and do the data support the conclusions?

Reviewer #1: Yes

Reviewer #2: Yes

2. Has the statistical analysis been performed appropriately and rigorously?

Reviewer #1: Yes

Reviewer #2: Yes

3. Have the authors made all data underlying the findings in their manuscript fully available?

Reviewer #1: Yes

Reviewer #2: Yes

4. Is the manuscript presented in an intelligible fashion and written in standard English?

Reviewer #1: No

Reviewer #2: Yes

The manuscript has been revised by a native English-speaking language professional editor

5. Review Comments to the Author

Reviewer #1: This study addresses continuous deep sedation until death (CDSUD) as practiced in the French overseas departments. The methodology consisted of surveying physicians who had completed the death certificates and comparing those who performed CDSUD with those who did not. The article is well written, clear, and concise. I have only two comments:

We thank you for assessing our paper and for your suggestions that we took into account in the revision of our manuscript

R1.1-The tables presenting only the weighted frequencies and weighted percentages are not straightforward to interpret. It would be more informative for the reader if a table with three columns were provided: the first including the unweighted counts and percentages (n), the second the weighted frequencies and weighted percentages (n ), and the third the p-value.

We changed the tables in order to comply with your proposal and displayed in tables both the unweighted and weighted frequencies and percentages. Supplementary file 7 with unweighted data was removed, as it is no longer useful.

R1.2-Considering the importance of the topic and the data collected in the overseas territories, it is regrettable that practitioners working in these territories were not involved in this publication. I am not convinced that physicians (VG and AE) from Nantes (mainland France), authors of this article, would have appreciated researchers from the overseas departments analyzing and using data from their hospital without being associated with the research. It even appears that physicians from La Réunion were approached to discuss the results. There are established teams and palliative care units in these territories, as well as (future) academics seeking to specialize in palliative care; excluding them from the study borders on disregard.

We understand the concerns you have raised regarding the principle, and we believe that the opportunity to get involved in studies should be given to locals. This research project and its protocol were adapted to overseas regions from those initially developed for mainland France, after a request from local health authorities. Local physicians were involved from preliminary discussions to first results presentations, without any of them showing interest in joining the research team, despite invitations to do so.

Reviewer #2: Overall assessment

This manuscript offers a valuable and original contribution by investigating continuous deep sedation until death (CDSUD) in the French overseas departments. The study addresses an important knowledge gap and provides new insights into the implementation of a unique legal framework. The design is thoughtful, and the results are of international interest.

That said, certain aspects require clarification or expansion before the paper can reach its full potential. These issues are not fundamental flaws, but they do limit the current strength of the conclusions.

We thank you for your thorough review of our paper and we have done our best to respond to all your comments/suggestions.

Major comments

R2.1 Response rate

The relatively low response rate (22.9%) is a key limitation. While this is common in physician surveys, some additional detail on representativeness (e.g., how respondents compared with the overall physician population in the overseas departments) would reassure readers about the reliability of prevalence estimates.

We cannot compare the responding physicians with the overall profile of physicians in these regions, since not all physicians are certifying doctors. Furthermore, the profile of certifying physicians is not a known statistic.

Additionally, while the respondents are physicians, the sample is actually a sample of deaths. The same physician may have responded for multiple deaths. If the certifying physicians were not the physician in charge of the deceased person, he/she could transfer the questionnaire to his/her colleague who was better able to respond.

However, we have clarified that we weighted the sample of responses to ensure representativeness with respect to all deaths during the period and mentioned that the weighting did not reveal any significant distortion between the distribution by sex, age, place of death, data collection period and overseas department.

We added the following paragraph in the methods section to replace the initial statements, in the statistical analysis section related to weighting process.

Representativeness of the responses

To ensure the representativeness of deaths, data were adjusted and standardised based on sex, age, place of death, data collection period, and overseas Department.

The differences between the responses and the total number of deaths over the period were minimal in terms of gender, age group and place of death. Unsurprisingly, they are slightly higher for later collection periods, probably because we were interviewing some of the same doctors. The main difference was between territories, with a higher response rate for La Réunion (23%) than for the other territories (15.4-19.7%). We used for the results the weighted frequencies and percentages to give a better representation of our initial population of deaths.

R2.2-Case classification

The adjudication process for CDSUD cases is innovative, but it introduces subjectivity. A clearer description of how disagreements were handled, or a basic measure of agreement between reviewers, would strengthen the credibility of this approach.

Thank you for your comment. We made the modifications in Supporting Material 4. We highlighted the agreement rate between evaluators and noted some elements to illustrate the changes.

R2.3-Clinical information

Because the study relies on death certificates, clinical details such as disease stage, comorbidities, and symptom burden are missing. Explicit acknowledgment of this limitation is needed, as it affects interpretation of appropriateness of CDSUD.

We added this information in the limitation section of the discussion (lines 414-416)

“We did not had variables on clinical details such as disease stage, comorbidities or symptom burden which may have limited our possibility to acknowledge the appropriateness of CDSUD”.

R2.4 - Collegial procedure

The finding that only about one-third of cases followed the formal collegial procedure is highly relevant. The discussion would benefit from exploring possible reasons for this gap (e.g., time constraints, institutional barriers, cultural norms) and its implications for policy and practice).

We altered the conclusion to introduce this idea (line 439-441)

“Discussion of the decision was frequent, although not always organised and formalised as a formal collegial procedure, indicating an area for improvement”.

R2.5-Cultural context

The overseas departments have distinctive cultural and religious environments, which likely influence end-of-life practices. A more developed reflection on this dimension would enrich the discussion and make the findings more interpretable for international readers.

We agree that cultural and religious contexts may influence end-of-life practices. We tried to addressed them in the manuscript, both in the contextual section (lines 84–85) and further discussed in the Discussion (line 337)."

Minor comments

R2.6 -Be

---

## [Decision Letter · Decision Letter 1]

10 Nov 2025

Dear Dr. Pennec,

Thank you for submitting your manuscript to PLOS ONE. After careful consideration, we feel that it has merit but does not fully meet PLOS ONE’s publication criteria as it currently stands. Therefore, we invite you to submit a revised version of the manuscript that addresses the points raised during the review process.

We look forward to receiving your revised manuscript.

Kind regards,

Stefaan Six, Ph.D.

Academic Editor

PLOS ONE

**Journal Requirements:**

**Additional Editor Comments:**

One reviewer notes that not all comments from the previous review round have been fully addressed. I invite the authors to do so, after which - subject to satisfactory revision - I am willing to consider the manuscript for acceptance.

Reviewers' comments:

Reviewer's Responses to Questions

**Comments to the Author**

Reviewer #1: (No Response)

Reviewer #2: All comments have been addressed

2. Is the manuscript technically sound, and do the data support the conclusions?

Reviewer #1: (No Response)

Reviewer #2: Yes

3. Has the statistical analysis been performed appropriately and rigorously?

Reviewer #1: Yes

Reviewer #2: Yes

4. Have the authors made all data underlying the findings in their manuscript fully available?

Reviewer #1: Yes

Reviewer #2: Yes

5. Is the manuscript presented in an intelligible fashion and written in standard English?

Reviewer #1: Yes

Reviewer #2: Yes

Reviewer #1: The authors have only partially addressed my previous comment regarding their limited familiarity with the French overseas territories (DOM) and the absence of collaboration with local practitioners. For example, they state that cancers are the leading cause of mortality (lines 85–86), which does not reflect the situation in the French West Indies, where cardiovascular diseases remain the main cause of death. Moreover, the French West Indies and Réunion have distinct clinical profiles and healthcare structures. Conducting additional analyses to differentiate findings between these two territories would strengthen the study and enhance its relevance for clinicians working in these regions.

Reviewer #2: (No Response)

**Do you want your identity to be public for this peer review?** For information about this choice, including consent withdrawal, please see our Privacy Policy

Reviewer #1: **Yes: ** Denis Boucaud-Maitre

Reviewer #2: No

---

## [Author Response · Author response to Decision Letter 2]

13 Nov 2025

Dear Editor,

Please find below our detailed responses to the comments raised by Reviewer 1. We have done our utmost to address each point. Regarding the suggestion to differentiate the results by territory, while we recognize its relevance, we were unfortunately unable to implement it due to data limitations and ethical constraints.

We hope that these revisions meet the reviewer’s expectations and look forward to your feedback.

Best regards,

The authors.

Dear Reviewer,

Reviewer #1:

We thank you for your review and the time you have dedicated to evaluating our manuscript. We have done our best to address all your comments and hope that the revisions now meet your expectations.

1.The authors have only partially addressed my previous comment regarding their limited familiarity with the French overseas territories (DOM) and the absence of collaboration with local practitioners.

We fully acknowledge the importance of involving local practitioners in research conducted in their territories, and we appreciate the opportunity to clarify our approach in response to your concerns.

The original protocol for this study was designed for mainland France, based on established methodologies used in other European countries. Following a request from one of the Regional Health Agencies of the French overseas territories, we adapted and extended the survey to these regions. Before initiating the study, we engaged in preliminary discussions with local physicians in one of the territories to assess the relevance and feasibility of the project. After data collection, we also presented our initial findings to local practitioners during dedicated meetings explicitly inviting them to participate in the research team if they wished to do so. While these exchanges did not result in any formal collaboration, we remain committed to fostering partnerships with local teams in future research.

2. For example, they state that cancers are the leading cause of mortality (lines 85–86), which does not reflect the situation in the French West Indies, where cardiovascular diseases remain the main cause of death.

We have checked this information. We wrote this sentence after analysing data from the centre of epidemiology of causes of deaths (CepiDc-Inserm) - the open data portal-. The table below gives the number of deaths by neoplasms and disease of circulatory system and the percentage per year and the figure gives the results by overseas departments.

During the last few years, the two main causes of deaths are neoplasms and diseases of circulatory systems, but neoplasms are now slightly above. To take into account your concern, we have altered the text:

Line 86: The French overseas departments operate under the same legal framework as mainland France and exhibit a similar distribution of causes of death. Over recent years, cancer has emerged as the leading cause of mortality, although cardiovascular disease persists at a high level—indeed higher than in mainland France [15].

Table 1 gives the number of deaths and percentage 2010-2013 of Guadeloupe, Martinique, French Guiana and La Réunion (both sexes, all ages)

Source: https://opendata-cepidc.inserm.fr/

Fouillet A, Aubineau Y, Godet F, Costemalle V, Coudin É., Grandes causes de mortalité en France en 2023 et tendances récentes, Bull Épidémiol Hebd., 2025;(13):218-243.

Godet F, Costemalle V, Aubineau Y, Fouillet A, Coudin É., Causes de décès en France en 2023 : des disparités territoriales, DREES, Études et Résultats n°1342

Figure R1: Number of deaths by neoplasms (blue) and disease of circulatory system (orange) by department.

3. Moreover, the French West Indies and Réunion have distinct clinical profiles and healthcare structures. Conducting additional analyses to differentiate findings between these two territories would strengthen the study and enhance its relevance for clinicians working in these regions.

We fully agree that the French West Indies and Réunion exhibit distinct clinical profiles and healthcare structures, despite sharing the same regulatory framework as mainland France. In our study, we already highlighted key structural differences, such as the higher proportion of deaths occurring at home, largely due to the lack of nursing home facilities in these territories.

In the discussion, we noted the significant differences in prevalence between the overseas departments, while exercising caution in interpreting these findings given the overall number of cases. Specifically, we reported a significantly higher prevalence of CDSUD in Réunion compared to the French West Indies and French Guiana. However, the limited number of CDSUD cases precluded a robust comparison of deceased patient or physician profiles by department, even when pooling data from the West Indies.

In the limitations section, we acknowledge that further territory-specific analyses—which would have been highly relevant for local practitioners—could not be performed due to data constraints. Unfortunately, the sample size did not allow for more detailed comparisons, despite the clear clinical interest of such an approach.

We have mentioned in the discussion that there were differences according to departments to emphasize this result.

Line 316: Although the prevalence rates differ significantly among the overseas departments, the overall number of cases warrants caution in interpreting these findings as indicative of differences in clinical practice.

And in the limits section, we acknowledge that we could not do more that we have done due to data limitations.

Line 400: While differences in prevalence between territories may exist, the limited number of cases precluded a territory-specific analysis.

---

## [Editor Report · Decision Letter 2]

16 Nov 2025

Deep and continuous sedation until death in the French overseas departments

PONE-D-25-49331R2

Dear Dr. Pennec,

We’re pleased to inform you that your manuscript has been judged scientifically suitable for publication and will be formally accepted for publication once it meets all outstanding technical requirements.

Kind regards,

Stefaan Six, Ph.D.

Academic Editor

PLOS ONE
---

## [Editor Report · Acceptance letter]

PONE-D-25-49331R2

PLOS ONE

Dear Dr. Pennec,

I'm pleased to inform you that your manuscript has been deemed suitable for publication in PLOS ONE. Congratulations! Your manuscript is now being handed over to our production team.

Kind regards,

on behalf of

Dr. Stefaan Six

Academic Editor

PLOS ONE